# Adalimumab-Induced Rhupus Syndrome in a Female Patient Affected with Anti-Citrullinated Protein Antibody (ACPA)-Positive Rheumatoid Arthritis (RA): A Case Report and Review of Literature

Ciro Manzo [1,*] and Alberto Castagna [2]

1   Internal and Geriatric Medicine Department, Rheumatologic Outpatient Clinic, Azienda Sanitaria Locale Napoli 3 Sud, Health District No. 59, Sant'Agnello, 80065 Naples, Italy
2   Primary Care Department, Casa Della Salute "Chiaravalle Centrale", Fragility Outpatient Clinic, Azienda Sanitaria Provinciale di Ca-tanzaro, Chiaravalle, 88064 Catanzaro, Italy; alberto.castagna@tiscali.it
*   Correspondence: manzoreumatologo@libero.it; Tel.: +39-081-533-1465

**Abstract:** We report a 38-year-old female patient affected with anti-citrullinated protein antibody (ACPA)-positive rheumatoid arthritis (RA) who developed mild hemolytic anemia (Hb = 10.5 vs. >12 gr/dL), indolent oral ulceration, ANA (1:1280, homogeneous pattern), and anti-dsDNA antibody positivity following 8 months of therapy with an adalimumab biosimilar (GP2017). Rhupus syndrome was diagnosed. Replacing GP2017 with infliximab, anemia, oral ulcer, and anti-dsDNA antibodies quickly disappeared, while low-titers (1:80) ANA are still present after more than a year. The possibility that the patient suffered from rhupus rather than drug-induced lupus erythematosus associated to anti-ACPA positivity RA was discussed. To date, after a 14-month follow-up, no manifestations of LE have reappeared. To the best of our knowledge, this is the first report of adalimumab-induced rhupus.

**Keywords:** adalimumab; biosimilars; rhupus syndrome; drug-induced lupus erythematosus

## 1. Introduction

In 1971, Schur first used the term "rhupus" to describe a syndrome in which lupus erythematosus (LE) and rheumatoid arthritis (RA) coexist in the same patient [1]. Rhupus syndrome can present as either LE or RA, but RA is the initial diagnosis in most patients. It is unusual that both diseases are simultaneously present. Its prevalence has been reported from 0.01 to 9.7%, depending on population setting and classification criteria; the median onset age is variable [2–4].

Adalimumab is the first fully human, high-affinity recombinant immunoglobulin G anti-tumor necrosis factor alfa (anti-TNF-$\alpha$) monoclonal antibody approved for the treatment of RA, and other autoimmune rheumatological and non-rheumatological diseases. In RA patients, its usual dosage is of 40 mg every two weeks by subcutaneous administration [5]. Adalimumab can trigger a drug induced LE (DILE) [6].

As of 2016, the European Medicine Agency (EMA) and the United States Food and Drug Administration (FDA) have approved seven adalimumab biosimilars. In line with the FDA and EMA regulatory requirements for biosimilar approval, the active substance of a biosimilar must be similar to the originator in both molecular and biological terms. Recently, a systematic review confirmed that adalimumab biosimilars have efficacy and safety comparable to the adalimumab reference product (so-called "originator") [7].

## 2. Case Report

In February 2020, we received a 38-year-old woman affected with ACPA-positive RA for consultation about anemia. RA had been diagnosed according to the criteria proposed

by the American Rheumatism Association in 1987 [8]. In particular, she had 6 of the proposed 7 criteria, namely: mornig stiffness lasting >1 h, arthritis of hand joints and symmetric arthritis of the wrist lasting >6 weeks, high serum rheumatoid factor (RF), and radiographic changes (Figures 1 and 2). In Table 1, we list the main laboratory data at the time of diagnosis of RA.

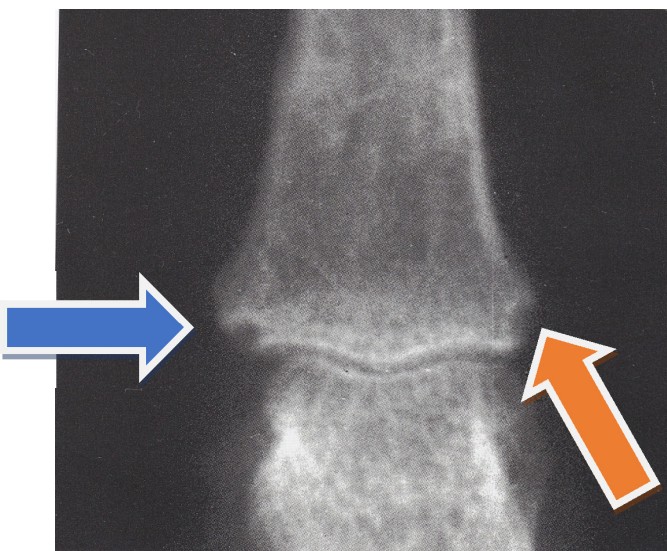

**Figure 1.** Proximal interphalangeal joint, third finger, right hand. Iuxta-articular erosions (see arrows).

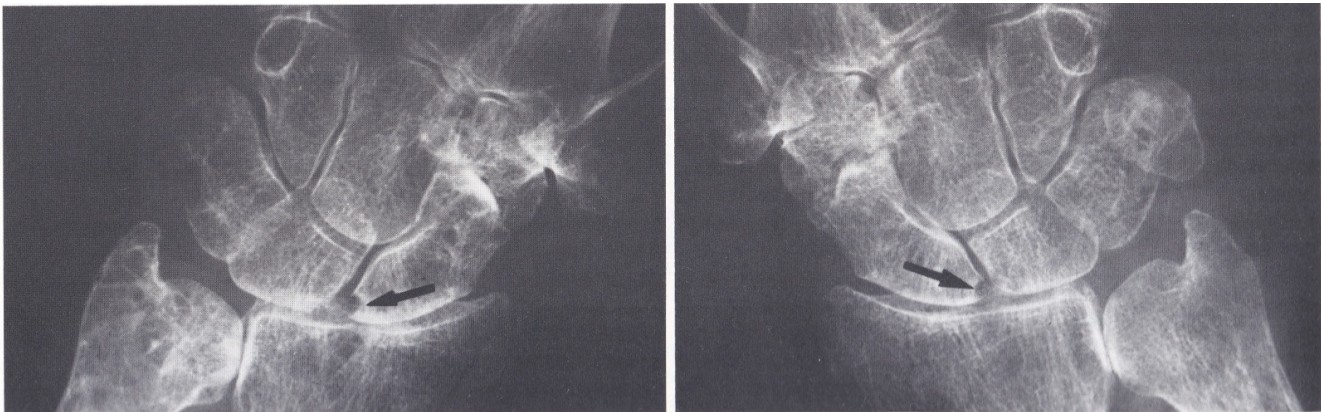

**Figure 2.** Bilateral and symmetrical erosions of the scaphoid bones.

**Table 1.** Main laboratory data at the time of RA diagnosis.

| |
| --- |
| ESR = 55 mm/h (n.v. < 15) |
| CRP concentrations = 15 mg/dL (n.v. < 0.3) |
| Hemoglobin = 12.2 gr/dL (n.v. > 12) |
| RF = 80 IU/mL (n.v. < 20) |
| ACPA = 200 IU/mL (n.v. < 18) |
| ANA < 1.40 |
| LAC, p-ANCA, c-ANCA: normal ranges |
| Renal and hepatic function tests: within their normal ranges |
| Occult blood testing in the stool: negative |
| Faecal calprotectin dosage: within normal range. |
| Hepatitis A, B and C serology: negative |

Abbreviations: ESR = erythrocyte sedimentation rate; CRP = C-reactive protein; n.v. = normal values; ACPA = anti-citrullinated protein antibodies; FR = rheumatoid factor; ANA = antinuclear antibodies; LAC = lupus anti-coagulant; ANCA = anti neutrophil cytoplasmic antibodies.

The patient had started therapy with adalimumab biosimilar (GP2017) 8 months prior, in addition to methotrexate (15 mg every week by subcutaneous administration), achieving a 28-joint Disease Activity Score with C-reactive protein [CRP]) concentrations (DAS28-CRP) of 2.90 (low disease activity according to Fransen et al. [9,10]).

According to laboratory findings, a mild autoimmune hemolytic anemia (AHA) was diagnosed. The main laboratory data at the time of our clinical examination were listed in Table 2. The patient refused genetic assessment.

**Table 2.** Main laboratory data at the time of our examination.

| |
|---|
| ESR = 38 mm/h (n.v. < 15) |
| CRP concentration = 1 mg/dL (n.v. < 0.3) |
| Hemoglobin = 10.5 gr/dL (n.v. > 12) |
| Reticulocyte count = 6.2 % (n.v. < 2.3%) |
| Total bilirubin = 1.8 mg/dL (n.v. < 1.0) |
| Indirect bilirubin = 1.2 mg/dL (n.v. < 0.6) |
| Iron = 193 mcg/dL (n.v. < 140) |
| Ferritin = 450 ng/mL (n.v. < 120)Haptoglobin = 300 mg/dL (n.v. < 150) |
| Direct Coombs test = positive |
| ACPA = 256.6 IU/mL (n.v. < 18) |
| ANA = 1:1280, homogeneous pattern (n.v. < 1:80) |
| Anti-dsDNA antibodies = 400 IU/mL (n.v. < 200) |
| Anti-phospholipid antibodies = negative |
| LAC = negative |
| Anti- beta2 glycoprotein 1 = negative |
| p-ANCA = negativec-ANCA =negative |
| Renal and hepatic function tests: within their normal ranges |

Abbreviations: ESR = erythrocyte sedimentation rate; CRP= C-reactive protein; n.v. = normal values; ACPA= anti-citrullinated protein antibodies; ANA = antinuclear antibodies; dsDNA = double-stranded DNA; LAC= lupus anti-coagulant; ANCA = anti neutrophil cytoplasmic antibodies.

During clinical examination, we noticed the presence of an oral ulcer localized to soft palate (Figure 3) that the patient had not reported because it was not sore.

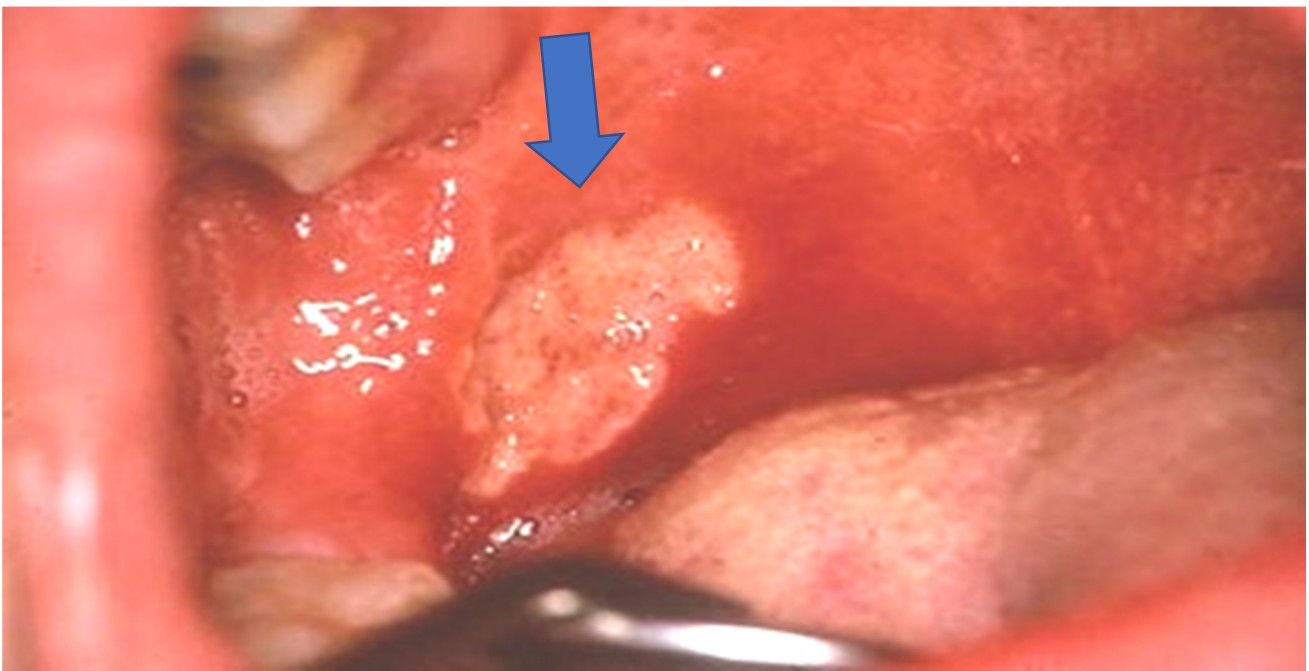

**Figure 3.** Oral ulcer (blue arrow).

The association of these clinical and laboratory findings (that are present in all classification criteria proposed for LE [11]) with ACPA+ RA authorized diagnosis of rhupus syndrome. Anemia progressively regressed after 12.5 mg/day prednisone, reappearing, however, when the daily dose of prednisone was reduced to 5 mg. At this point, we proposed the patient replace adalimumab with another ant-TNF-α (infliximab). Indeed, according to data we found in the literature [7], we hypothesized that adalimumab could have triggered manifestations of LE. Prednisone was stopped; methotrexate was continued. After one month, hemolytic anemia and oral ulcer had disappeared. To date, hemolytic anemia, oral ulcer, and anti-dsDNA are absent; ANA titer is 1:80 (homogeneous pattern); DAS28-CRP is 2.47.

## 3. Discussion

In our patient affected with RA, mild hemolytic anemia, indolent oral ulcer, ANA, and anti-dsDNA positivity followed 8-month therapy with an adalimumab biosimilar.

Did the patient suffer with DILE [12,13] associated with RA or with rhupus?

To date, according to the World Health Organization pharmacovigilance database (WHO VigiBase), anti-TNF-α are the molecules that most commonly trigger DILE [14]. Adalimumab can induce DILE in patients with RA.

We performed a literature search in Embase and PubMed (OVID interface) databases using the key terms "rhupus", "adalimumab", "adalimumab-induced rhupus", "adalimumab reference product", "adalimumab originator", "adalimumab biosimilars", "drug-induced lupus erythematosus", "lupus-like syndrome", "rhupus treatment", and "rhupus review". A safety analysis in global clinical trials and US post-marketing surveillance of patients with RA found a DILE incidence of between 0.08 and 0.10% [15]. More recently, in a 10-year study that included 12,345 RA patients treated with adalimumab, a lupus-like syndrome was reported only in 0.20% (0.07% as serious events) of patients [16]. Some working hypotheses have been proposed to explain the onset of LE-like manifestations in patients treated with anti-TNF- α: for instance, the "cytokine shift" from Th1 to Th2, interleukin 10 (IL-10) and interferon-alfa, leading to the production of autoantibodies; or an inhibition of cytoxic T-cells, able to reduce the elimination of autoantibody-producing B-cells [12].

According to some investigators [17], the presence of ACPA, high CRP concentrations, and shared epitope [18] must be present in a patient with coexistent RA and LE clinical and laboratory criteria for the diagnosis of rhupus to be proposed. Our patient suffered from ACPA-positive RA and had increased CRP concentrations at the time of LE manifestations. As already mentioned, she refused a genetic assessment. As a matter of fact, joint damage (erosive or not) alone may have no significant value in the differential diagnosis between RA and LE [3,19]. Furthermore, CRP concentrations do not increase during flares of LE [20]. Finally, it is common knowledge that the most common autoimmune manifestation associated with anti-TNF therapy is the development of ANA and ds-DNA autoantibodies. ANA and anti-dsDNA antibodies can be present both in DILE and in rhupus, and it is common knowledge that ANA positivity can also persist for a long time after drug withdrawal.

The possibility that AHA could be a crosswalk between RA and LE has already been reported [3,21,22]. Recently, some investigators reported a 29-year-old woman in whom AHA was the first manifestation of rhupus [23], as in our case report. The incidence of AHA in RA has been described in 2.1–2.5% [24]. Among 56 patients with rhupus, the prevalence of AHA was just 5% versus 21% in LE patients [3]. Other series report no cases of rhupus and AHA [25].

In general, rhupus patients may present with any of the systemic manifestations of LE. However, a multicentric comparative study on clinical and serological characteristics between patients with rhupus and those with LE and RA reported that the main LE-like clinical manifestations in rhupus patients were cutaneous manifestations (oral ulcerations are among these), hematological manifestations, and serositis [26], so confirming data

already reported by other investigators [3]. It is worth pointing out that these two studies had by far the most numerous casuistry.

Efficacy and safety of anti-TNF-$\alpha$ has rarely been assessed in patients with rhupus. Interestingly, in an open-label study involving 15 patients following anti-TNF-$\alpha$ treatment, the authors observed no lupus flare in the three rhupus patients treated with adalimumab originator [27]. This was the only study we found in the literature that focused on efficacy and safety of adalimumab in patients affected with rhupus. We found no data regarding efficacy and safety of adalimumab biosimilars and infliximab in patients with rhupus.

Finally, the fact that—in our patient—the switch from adalimumab to infliximab allowed resolution of rhupus syndrome deserves a short discussion. Both adalimumab and infliximab are members of the same class of drugs, that is the anti-TNF-$\alpha$ or TNF $\alpha$ blockers. The possibility that these molecules may be associated with different efficacy and adverse events [28] is the basis for switching from one anti-TNF-$\alpha$ to another [29]. Different structural biology [30] and genetic and/or epigenetic variation [31] may play a crucial role. To date, however, the pharmacogenomic bases for stratifying RA patients according to anti-TNF treatment are still discussed [32].

## 4. Conclusions

In conclusion, we report a case of adalimumab-induced rhupus, whose resolution was made possible by replacing adalimumab with a different anti-TNF-$\alpha$ such as infliximab. To date, after a 14-month follow-up, no other manifestations of LE have appeared. To the best of our knowledge, this is the first report of adalimumab-induced rhupus.

**Author Contributions:** All authors contributed equally to the article. All authors have read and agreed to the published version of the manuscript.

**Funding:** This research received no external funding.

**Institutional Review Board Statement:** The study was conducted according to the guidelines of the Declaration of Helsinki and approved by the Ethics Committee of Health District no. 59 (protocol code: 0001/59/Na3sud, 21 March 2021).

**Informed Consent Statement:** Informed consent was obtained from the patient involved in the study. Written informed consent has been obtained from the patient to publish this paper.

**Conflicts of Interest:** The authors declare no conflict of interest.

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
