# Peer review of "Adalimumab-Induced Rhupus Syndrome in a Female Patient Affected with Anti-Citrullinated Protein Antibody (ACPA)-Positive Rheumatoid Arthritis (RA): A Case Report and Review of Literature"

_clinpract, doi:10.3390/clinpract11030055_

Round 1

Reviewer 1 Report

1. The initial clinical presenatation of the patient with RA shuld be described.

2. Image of bilateral hands or other involed peripheral joints related bony erosion should be presented

3. The initial laboratory data of the patient should be shown

4. The medication of TNFI may be associated with immune dysregulation in the patient with RA, and adalimumab or infliximab ia a member of TNFI, and different immune response in these two types of TNFI, why? This difference should be discussed

Author Response

Dear Reviewer,

we thank You so much for Your valuable comments and suggestions.

Please, read the attached file.

With kindest regards.

Reviewer 2 Report

Dear Authors,

Please extent your introduction with more references

Please add more figures with high resolution

Please add more conclusions

Author Response

Dear Reviewer,

thank You !

You can read our point-by-point answers in the attached file.

Reviewer 3 Report

This is a good paper about Adalimumab-induced rhupus syndrome in a female patient affected with anti-citrullinated protein antibodies (ACPA)-positive rheumatoid arthritis. Let me ask that why is the mechanism happened. For example why infliximab does not occured?

Author Response

Dear Reviewer,

we wish to thank You for Your time and attention.

Your questions are useful discussion points , and we answered them in the newer Discussion section of our manuscript.  Please, read what we highlighted in Yellow.

We hope that Your comments were satisfactorily met.

We are sure that quality of presentation and scientific soundness of our case report have improved.

Thanks, again !

Reviewer 4 Report

The authors described a patients with RA (ACPA+) in which adalimumab triggered the development of SLE.

This is a well-known association, usually patients develop lupus-like features but rarely even a full-blown SLE can occur. Thus, the case is not novel and does not add anything to current knowledge.

Author Response

Dear Reviewer,

we are very sorry for the Your unfair comments.

Indeed, we reported a patients with ACPA+ RA in which adalimumab triggered the development of rhupus syndrome.

There are....some differences between rhupus syndrome and a drug-induced SLE, and You should know them.

In our Discussion section, we clearly addressed this point.  Maybe You read our manuscript hastily?

It should be noted that none of the other three reviewers raised such a rebuttal. 

Ciro Manzo and Alberto Castagna

Round 2

Reviewer 2 Report

Dear authors,

Congratulation. Now it is ok

Reviewer 3 Report

OK, this paper is well modified.

Reviewer 4 Report

as already said, this is not a novel case.